# Enhancing Endocannabinoid Signaling via β-Catenin in the Nucleus Accumbens Attenuates PTSD- and Depression-like Behavior of Male Rats

**DOI:** 10.3390/biomedicines10081789

**Published:** 2022-07-25

**Authors:** Tomer Mizrachi Zer-Aviv, Larglinda Islami, Peter J. Hamilton, Eric M. Parise, Eric J. Nestler, Brenda Sbarski, Irit Akirav

**Affiliations:** 1Department of Psychology, School of Psychological Sciences, and the Integrated Brain and Behavior Research Center, University of Haifa, Haifa 3498838, Israel; tomer.mizrachi6@gmail.com (T.M.Z.-A.); laislami@uni-mainz.de (L.I.); bsbarski@gmail.com (B.S.); 2Nash Family Department of Neuroscience and Friedman Brain Institute, Icahn School of Medicine at Mount Sinai, New York, NY 10029, USA; peter.hamilton@vcuhealth.org (P.J.H.); eric.parise@mssm.edu (E.M.P.); eric.nestler@mssm.edu (E.J.N.)

**Keywords:** rat, post-traumatic stress disorder (PTSD), cannabinoids, URB597, mGluR5, CB1 receptor, fatty acid amide hydrolase (FAAH), β-catenin

## Abstract

Inhibition of fatty acid amide hydrolase (FAAH), which increases anandamide levels, has been suggested as a potential treatment for stress-related conditions. We examined whether the stress-preventing effects of the FAAH inhibitor URB597 on behavior are mediated via β-catenin in the nucleus accumbens (NAc). Male rats were exposed to the shock and reminders model of PTSD and then treated with URB597 (0.4 mg/kg; i.p.). They were tested for anxiety- (freezing, startle response), depression-like behaviors (despair, social preference, anhedonia), and memory function (T-maze, social recognition). We also tested the involvement of the CB1 receptor (CB1r), β-catenin, and metabotropic glutamate receptor subtype 5 (mGluR5) proteins. URB597 prevented the shock- and reminders-induced increase in anxiety- and depressive-like behaviors, as well as the impaired memory via the CB1r-dependent mechanism. In the NAc, viral-mediated β-catenin overexpression restored the behavior of rats exposed to stress and normalized the alterations in protein levels in the NAc and the prefrontal cortex. Importantly, when NAc β-catenin levels were downregulated by viral-mediated gene transfer, the therapeutic-like effects of URB597 were blocked. We suggest a potentially novel mechanism for the therapeutic-like effects of FAAH inhibition that is dependent on β-catenin activation in the NAc in a PTSD rat model.

## 1. Introduction

Stressful life events can substantially impact the brain and induce psychiatric disorders such as post-traumatic stress disorder (PTSD) and depression [1].

A large body of evidence, including our own findings, points to the endocannabinoid (ECB) system as a possible therapeutic target to help treat PTSD [2,3,4,5,6,7,8,9,10,11,12,13,14,15]. The ECB system contains the cannabinoid receptors CB1 (CB1r) and CB2 (CB2r); endogenous ligands: N-arachidonyl ethanolamine (anandamide; AEA) and 2-arachidonoyl glycerol (2-AG); and their degrading enzymes: 2 monoacylglycerol lipase (MAGL) for 2-AG and fatty acid amide hydrolase (FAAH) for AEA.

Using a rat model for PTSD in an inhibitory avoidance apparatus, wherein rats are exposed to a single severe foot shock followed by exposure to contextual reminders, we found that acute administration of the FAAH inhibitor URB597 or the CB1/CB2 receptor agonist WIN55,212-2 after shock exposure prevented anxiety- and depression-like behaviors [3,15,16,17]. Unlike direct cannabinoid agonists, URB597 does not cause classical cannabinoid side effects such as catalepsy, hypothermia, hyperphagia, and abuse potential [18,19]. However, the mechanism for the therapeutic-like effects of URB597 requires further investigation.

The Wnt/β-catenin pathway was found to play a significant role in anxiety and depressive symptoms [20,21] and to regulate pro-resilient and anxiolytic-like effects in the nucleus accumbens (NAc) by activating a network that includes Dicer1 and downstream microRNAs [22]. Moreover, overexpressing GSK-3β (i.e., the kinase which phosphorylates β-catenin) in the NAc induced depressive-like behavior, whereas GSK-3β inhibition promoted resilience [23]. Similarly, contextual and cued fear memory increased GSK-3β phosphorylation in the hippocampus and amygdala [21,24,25]. We have previously demonstrated that enhanced extinction kinetics in rats exposed to shock and reminders was significantly associated with increased expression of β-catenin in the NAc; hence, suggesting that the expression of β-catenin in the NAc is linked with a resilient response to the stressor [26]. When β-catenin levels in the NAc were inhibited using the non-selective β-catenin antagonist sulindac, it blocked the therapeutic-like effects of WIN55,212-2 on extinction [26]. The involvement of the Wnt/β-catenin pathway in antidepressant and pro-cognitive effects also has been observed in the medial prefrontal cortex (mPFC) [27,28].

In vivo and in vitro evidence suggests that Wnt/β-catenin signaling is downstream of metabotropic glutamate receptor subtype 5 (mGluR5); hence, mGluR5 signaling plays a key role in controlling neuronal gene expression by regulating the assembly of the N-cadherin/β-catenin complex and consequently the expression of REST/NRSF (Repressor element 1-silencing transcription factor/neuron-restrictive silencer factor) in primary corticostriatal neurons [29]. Importantly, mGluR5 is implicated in the pathophysiology of several psychiatric disorders [30]. The direction of mGluR5 modulation that elicits antidepressant/anxiolytic-like effects has been inconsistent across studies [31,32,33]. However, the PFC of postmortem brains of major depressive disorder patients shows reduced mGluR5 protein expression [34]; in rodents, an essential role for mGluR5 in the NAc was found in promoting stress resilience, suggesting that a deficit in mGluR5-mediated signaling in this region may represent an endophenotype for stress-induced depression [35]. In a recent study, NAc mGluR5 activation ameliorated the effects of stress on depression-like behavior and pain, through ECB mediation, suggesting an association between ECB signaling and the expression of mGluR5 in stressed rats [36].

The aim of the current study is to examine the role of β-catenin in the stress-attenuating effects of FAAH inhibition and to assess its function in anxiety- (freezing, startle response), and depression-like behaviors (forced swim test, social preference, saccharin preference), as well as memory function (water T-maze, social recognition). To that end, we overexpressed and downregulated β-catenin in the NAc by viral-mediated gene transfer and assessed the effects of URB597 in these relevant stress-related behaviors and several proteins of interest (i.e., β-catenin, mGluR5 and CB1r) in the NAc, PFC, hippocampus and amygdala. We hypothesized that URB597 would prevent the shock- and reminders-induced alterations in behavior, and that these therapeutic-like effects of URB597 would be mediated via β-catenin in the NAc. 

## 2. Materials and Methods

### 2.1. Subjects

Male Sprague Dawley rats (60 days old, ~225 g; Envigo, Jerusalem, Israel) were caged together according to their treatment group (4 per cage; 59 × 28 × 20 cm) at 22 ± 2 °C under 12 h light/dark cycles. Plastic hoses were placed in each cage to enrich the animals’ environment. For the saccharin preference test, each rat was placed in a separate cage. Rats were allowed water and laboratory rodent food ad libitum. 

### 2.2. Drug Treatment

The FAAH inhibitor URB597 (0.4 mg/kg, i.p.) and the CB1r antagonist AM251 (0.3 mg/kg, i.p.) (Cayman Chemicals, Ann Arbor, MI, USA) were dissolved in dimethylsulfoxide 5%, Tween-80 and saline 90% (Sigma-Aldrich, Rehovot, Israel). Drug doses are based on previous findings [3,4,16,37]. 

### 2.3. Shock and Situational Reminders

Rats were exposed to a single shock in a passive avoidance apparatus divided into two equal-size compartments (one light and one dark), which were separated by a guillotine door [15,17]. On shock day, rats were placed in the light compartment; once the rat entered the dark compartment, a single foot shock (1.5 mA, 10 s) was delivered (Coulbourn Instruments, Whitehall, PA, USA). The no-shock groups received the same treatment, but with the shock mechanism inactivated.

For situational reminders (SRs), a rat was placed in the lighted start chamber for 60 s, with the guillotine door closed to prevent the rat from entering the shock compartment (to avoid extinction). No further shocks were administered. We used a video camera to monitor the duration of freezing behavior during the 60 s SR in the lighted chamber. Freezing was measured before shock administration (baseline) and during the five days of SRs (SR1-SR5). The percentage of changed pixels between two adjacent 1 s images was calculated, and if the percentage of change in images was <0.05%, the rat was scored as “freezing” [38]. Freezing was defined as the absence of all movement except for respiration [39]. 

### 2.4. Behavioral Testing

All rats were tested in all behavioral procedures in the same order. The most aversive test (forced swim test—FST) was last. Tests were separated by a 24 h period. All behavioral tests were conducted under dim lighting (15–20 lx) and took place between 12:00 and 16:00. Behavioral testing was taken after SR5 in order to examine the long-term effects of the traumatic event. In a previous study, we found that shocked rats exposed to SRs persistently avoided the dark chamber, whereas, shocked rats not exposed to SRs demonstrated increased avoidance on the first extinction trial, but their extinction kinetic was intact [7]. 

### 2.5. Acoustic Startle Response (ASR)

A soundproof chamber (25 × 25 × 25 cm) containing an acrylic animal holder (8 × 8 × 16 cm; Coulbourn Instruments, Whitehall, PA, USA) that is connected to a piezoelectric accelerometer. For consistency between chambers and experiments, calibration of sensitivity to the movement and sound levels was applied. A single rat was placed in the holder for a 5 min acclimatization period; 30 acoustic startle trials (98 or 120 dB; 50 ms duration, 20–40 s intertrial interval) were presented over the 68 dB white noise background. We analyzed the mean startle amplitude which indicates the averaged response to the 98 and 120 dB in mV [40]. 

### 2.6. Social Preference and Social Recognition

This task assesses sociability and short-term social memory. Habituation for 1 h to the transparent corrals in the home cages of the juvenile and experimental rats preceded the training. The object and the juvenile rat were confined to the corrals that were placed 10 cm from the two opposite corners of the arena. The corrals (9 cm in diameter) allowed physical contact with the experimental rat [3]. The objects were children’s Lego blocks.

For the preference and recognition phases, the experimental rat was given 5 min exploration with an intertrial interval of 30 min in a holding cage. For preference, the rat explored the novel object and the unfamiliar juvenile. The ratio was calculated as the time spent exploring the juvenile rat divided by the total exploration time (juvenile rat + novel object). For recognition, the rat explored a novel juvenile and the familiarized juvenile. The ratio was calculated as the time spent exploring the novel juvenile divided by total exploration time (familiar + novel juveniles). The trials were videotaped (Logiteck, C922 Pro Stream, Newark, CA, USA) and recorded (NCH Software, Greenwood Village, CO, USA) and the time spent in corral exploration was analyzed using EthoVision XT9 software.

### 2.7. Water T-Maze (WTM)

The WTM is a black Plexiglas maze (length of stem, 64 cm; length of arms, 43 cm; width, 13 cm; height of walls, 42 cm). A transparent plastic escape platform (12 × 12 cm) is hidden at the end of one of the two arms that are filled with water (23 ± 1 °C). The test has two phases: acquisition and reversal. 

For the first acquisition phase, a rat needs to choose the right arm out of two. If it chooses the right arm, it remains on the platform for 10 more seconds. If it chooses the wrong arm, the rat is confined to the arm for 20 s. 

In the second reversal phase (performed 24 h after the acquisition phase), each rat was first tested in a probe trial without the platform. Once the rat entered an arm, it was removed from the maze. If the non-reinforced arm was chosen, the rat was retrained on the discrimination of the previous day and tested again 24 h later. If the reinforced arm was chosen, the rat was trained on the reversal of that discrimination, i.e., the platform was moved to the opposite arm [41]. In both phases, the number of correct trials was recorded until the rat reached five consecutive correct trials. 

### 2.8. Forced Swim Test (FST)

This test is based on the assumption that when a rat is placed in a container filled with water, it will at first make efforts to escape but eventually will exhibit immobility that may reflect despair [42]. An acrylic cylindrical container (62 cm diameter, 40 cm height) was filled with water at a temperature of 24 °C. The water level (34 cm) was such that a rat could not touch the bottom with its hind paws. A rat was forced to swim for 15 min inside the container. Then, after 24 h in the home cage, the rat was put back in the container and forced to swim for 5 min. The time spent immobile was recorded. Immobility was defined as the lack of motion, except for movements necessary to keep the rat’s head above water [43,44].

### 2.9. Saccharin Preference Test

Water bottles were removed before the dark part of the cycle and replaced with two bottles, one filled with water and the other with saccharin (0.01 mg/L, dissolved in water, Sigma-Aldrich, Rehovot, Israel). Saccharin consumption was measured during the 12 dark hours of the cycle. Following one night of habituation to saccharin, the saccharin preference ratio was calculated as the saccharin consumption divided by the total consumption (consumption of saccharin/water + saccharin).

### 2.10. Western Blotting (WB)

Rats were euthanized and brain tissues of the mPFC (prelimbic and infralimbic punched together), NAc shell, basolateral amygdala (BLA), and CA1 area of the hippocampus were removed by cryostat using a 0.5 mm puncher (coordinates relative to Bregma in mm: mPFC: anteroposterior (AP), +3.72; medial–lateral (ML), ±0.4; ventral (V), −4.8; NAc shell: AP, +1.6; ML, ±1; V, −5.5; BLA: AP, −1.596; ML, ±4.2; V, 8.45; CA1: AP, −4.2; ML, ±2.5; V: −2.5). Protein levels were determined by the bicinchoninic acid protein assay kit (Thermo Fisher Scientific, Waltham, MA, USA). The samples were then diluted in an SDS sample buffer, boiled for 5 min at 100 °C, and then stored at −80 °C. On running day, wells were loaded with 30 μL of samples. Each gel contained at least one sample from each group. Aliquots were subjected to SDS–PAGE (7.5% polyacrylamide; Sigma-Aldrich, Rehovot, Israel) and immunoblot analysis.

Blots were incubated with CB1r (1:1000; abcam, Cambridge, UK; ab25932 [ERP23934-20]), β–catenin (1:5000; abcam, Cambridge, UK; ab32572 [E247]), or mGluR5 (1:5000; abcam, Cambridge, UK; ab76316 [ERP2425Y]) antibodies overnight at 4 °C. This was followed by a 1 h incubation with an HRP-linked secondary antibody at room temperature (1:10,000; goat anti-rabbit IgG; Jackson ImmunoResearch Laboratories, West Grove, PA, USA; 111-035-144). Blots were visualized with ECL (Bio-Lab, Jerusalem, Israel) and an XRS charge-coupled device camera (BioRad Laboratories, Hercules, CA, USA). We used the Quantity One software (BioRad Laboratories, Hercules, CA, USA) to assess blot density. All protein samples were standardized with β-actin (1:1000; Cell Signaling, Danvers, MA, USA; #5125 [13E5]). For antibody specificity, see the Appendix A.

### 2.11. Viral-Mediated Gene Transfer

The replication-deficient *herpes simplex virus* (HSV) p1005 vector is a “short-term” vector, derived from herpes simplex virus-1 with a high titer range (3–5 × 10^8^ transduction unit, TU/mL; an illustration of a modified HSV amplicon plasmid is presented in the Appendix A). Stereotactic surgery was performed on rats under anesthesia of Domitor (2%, 10 mg/kg, i.p.) and ketamine (10%, 100 mg/kg, s.c.) (Vetmarket, Modiin, Israel). A total of 1μL of the HSV viral vector or green fluorescent protein (GFP) was infused bilaterally into the NAc (Stoelting, Wood Dale, IL, USA) at a rate of 0.1 μL/min (coordinates relative to Bregma: AP, +1.6 mm; LM, ±1 mm; V, −5.5 mm). Vectors were used to overexpress (OE) or downregulate (DR) the expression of β-catenin compared to a GFP control five days before shock day; the vector is expressed in vivo within 2–3 h, with maximal expression from 3–5 days post-injection that lasts only 8 days in vivo [23,45,46,47]. The viral dose was determined by rendering the >90% cell infection rate in brain tissue, diluted in 60% PBS. 

The needle was held in place for 5 additional minutes before being slowly withdrawn. Animals were allowed 5 days of recovery before behavioral experiments began. 

### 2.12. Perfusion and Immunohistochemistry (IHC)

Perfusion:

In anesthetized rats (Domitor and ketamine), brains were perfused with 4% paraformaldehyde solution (Santa Cruz Biochemicals, Dallas, TX, USA). Post-fixation brains were kept at −80 °C [48].

GFP detection:

Brains were sectioned in 35-μm-thick slices using cryostat microtome (Leica Biosystems, Deer Park, IL, USA) and stored at 4 °C in PBS. Then, slices were washed three times for 15 min each in 1 × PBS (Sigma-Aldrich, St. Louis, MO, USA). After the washing procedure, the brain slices were mounted on super frost glass slides using PBS as a mounting solution and left to dry for 24 h. Glass slides were then stored at 4 °C in a dark chamber. Staining was documented using a confocal microscope at 5×, 10×, and 40× zoom (ZEISS, Jena, Germany).

### 2.13. Experimental Design

a. Study design for experiments 1 and 2: on day 0, male rats were exposed to a single, severe foot shock (1.5 mA, 10 s) in an inhibitory avoidance apparatus, followed by exposure to contextual 1 min situational reminders (SRs) of the shock on days 5, 10, 15, 20, and 25 (Figure 1a). Freezing was measured during exposure to SR. Drugs (Vehicle, URB597, AM251, AM251 + URB597) were administered i.p. 1 h after shock exposure. Following habituation to saccharin on day −3 (sacc’ hab; pre-shock), preference was tested on day −2 pre-shock and on days 2, 7, and 14 post-shock. Acoustic startle response (ASR) testing was administered twice: one day before the shock (−1; ASR1) and one day after the last reminder (day 26; ASR2). On day 27, rats were exposed to the social preference and recognition tests, on days 28–29 to the water T-maze (WTM), and on days 30–31 to the forced swim test (FST). In experiment 1, four groups (n = 10 for all groups) were administered with vehicle or URB597, and brains were removed on day 32 for western blotting (WB). In experiment 2, six groups (n = 8 for all groups) were administered with vehicle, AM251, or AM 251 + URB597. No rats were excluded from the experiments.

b. Study design for experiments 3 and 4: on day −5, the herpes simplex virus (HSV) vector was injected bilaterally into the nucleus accumbens (NAc) to overexpress (OE) or downregulate (DR) β-catenin (Figure 1b). On day 0, male rats were exposed to a single severe foot shock (1.5 mA, 10 s) in an inhibitory avoidance apparatus followed by exposure to contextual 1 min situational reminders (SRs) of the shock on days: 2, 5, 8, 11, and 14. Following habituation to saccharin on day −3 (sacc’ hab; pre-shock), preference was tested on days −2 pre-shock and on days 2, 7, and 14 post-shock. Acoustic startle response (ASR) testing was administered before the shock (day 1; ASR1) and one day after the last reminder (day 15; ASR2). In experiment 4, drugs (Vehicle, URB59751) were administered i.p. 1 h after shock exposure. On day 16 rats were exposed to the social preference and recognition tests, on days 17–18 to the water T-maze (WTM), and on days 19–20 to the forced swim test (FST). For experiment 3, the brains were removed on day 21, and β-catenin, mGluR5, and CB1r expression were measured using western blotting (WB). In experiment 3, GFP or OE was delivered to four groups (n = 7–10 for all groups). In experiment 4, GFP or DR was delivered to eight groups (n = 8 for all groups). No rats were excluded from the experiments.

### 2.14. Statistical Analysis

The results are expressed as means ± SEM. For statistical analysis, we used one-way ANOVAs, two-way ANOVAs, repeated measures ANOVAs, t-tests, and Pearson bivariate correlation tests, as indicated. All post hoc comparisons were made using Tukey’s range test. Significance was chosen at *p* ≤ 0.05. Data were analyzed using SPSS 27 (IBM, Chicago, IL, USA). Normality assumption was examined using the Kolmogorov–Smirnov and Shapiro–Wilk tests.

## 3. Results

### 3.1. Experiment 1: The Effects of URB597 on Behavior and the Expression of β-Catenin in Rats Exposed to Shock and Reminders

Rats were exposed to a single severe foot shock followed by situational reminders. Drugs were administered i.p. 1 h after shock exposure, and then the rats performed a battery of behavioral tests; brains were removed after 24 h and β-catenin expression was measured (for detailed study design, see Section 2.13 Experimental Design).

#### 3.1.1. Freezing

Freezing behavior was monitored during the 1 min exposure to the SR conditions (baseline and SR1-SR5) (Figure 1a). Repeated measures ANOVA (shock × drug × SR; 2 × 2 × 6) revealed significant main effects of drug [F(1,36) = 37.737, *p* < 0.001], shock [F(1,36) = 78.650, *p* < 0.001], SR day [F(5,180) = 11.773, *p* < 0.001]. We also detected the following interactions: shock × drug [F(1,36) = 36.301, *p* < 0.001], shock × SR [F(5,180) = 10.960, *p* < 0.001], drug × SR [F(5,180) = 4.910, *p* < 0.001], and shock × drug × SR [F(5,180) = 4.823, *p* < 0.001]. Post hoc analysis revealed that on SR1 to SR5 the Shock/Veh group demonstrated a significant increase in freezing levels compared with the NoShock/Veh group (*p* < 0.001) and the Shock/URB group (SR1; SR2; SR4; SR5: *p* < 0.01; SR3: *p* < 0.001). This suggests that URB597 prevented the shock-induced increase in freezing behavior. In addition, the Shock/URB group showed increased freezing levels compared with the NoShock/URB group (SR1: *p* < 0.001; SR2; SR3: *p* < 0.01; SR4: *p* < 0.05).

#### 3.1.2. Saccharin Preference 

Saccharin preference was tested on day 2 pre-shock and on days 2, 7, and 14 post-shock (Figure 1b). Repeated measures ANOVA (shock × drug × day; 2 × 2 × 4) revealed significant effects of drug [F(1,36) = 16.109, *p* < 0.001] and day [F(3,108) = 5.633, *p* < 0.01]; with the following interactions: shock × drug [F(1,36) = 8.970, *p* < 0.01], shock × day [F(3,108) = 2.896, *p* < 0.05], drug × day [F(3,108) = 5.723, *p* < 0.01], shock × drug × day [F(3,108) = 6.052, *p* < 0.01]. Post hoc analysis revealed that on days 2 and 7, the Shock/Veh group demonstrated decreased saccharin preference compared with the NoShock/Veh (day 2: *p* < 0.001; day 7: *p* < 0.01) and Shock/URB (*p* < 0.001) groups. Hence, URB597 prevented the shock and reminders induced a decrease in saccharin preference (i.e., anhedonia).

#### 3.1.3. Acoustic Startle Response 

The ASR test was performed on days 1 (pre-shock) and 26 (Figure 1c). We performed a repeated measures ANOVA (shock × drug × time; 2 × 2 × 2) on mean amplitude, and identified significant effects of shock [F(1,36) = 4.060, *p* < 0.05] and time [F(1,36) = 7.799, *p* < 0.01], with the following significant interactions: shock × time [F(1,36) = 5.3, *p* < 0.05], drug × time [F(1,36) = 6.779, *p* < 0.05], and shock × drug × time [F(1,36) = 8.646, *p* < 0.01]. Post hoc analysis on ASR2 revealed a significant increase in amplitude in Shock/Veh compared with Shock/URB and NoShock/Veh (*p* < 0.05 in both cases). This suggests that URB597 restored the shock/reminders-induced increase in startle response.

#### 3.1.4. Social Tests 

We performed the social tests on day 27 (Figure 1d). We ran a two-way ANOVA (shock × drug; 2 × 2) and found significant effects of shock [preference: F(1,36) = 4.537, *p* < 0.05; recognition: F(1,36) = 23.199, *p* < 0.001] and drug [preference: F(1,36) = 22.928, *p* < 0.001; recognition: F(1,36) = 6.619, *p* < 0.05], with a significant drug × shock interaction [preference: F(1,36) = 6.259, *p* < 0.05; recognition: F(1,36) = 5.905, *p* < 0.05]. Post hoc analysis revealed a significant decrease in the exploration ratio in both tasks in the Shock/Veh group compared with the NoShock/Veh group (preference: *p* < 0.01; recognition: *p* < 0.001) and Shock/URB597 group (*p* < 0.001 for both preference and recognition). Hence, URB597 restored the impairing effects of shock and reminders on social behavior. For total exploration time, see the Appendix A.

#### 3.1.5. Water T-Maze 

For the WTM, tested on days 28–29 (Figure 1e), a repeated measures ANOVA (shock × drug × test; 2 × 2 × 2) revealed significant effects of shock [F(1,35) = 60.717, *p* < 0.001], drug [F(1,35) = 77.881, *p* < 0.001] and test [F(1,35) = 13.271, *p* < 0.01], with a significant shock × drug interaction [F(1,36) = 53.689, *p* < 0.01]. Post hoc analysis revealed that the Shock/Veh group showed a significant increase in the number of trials required to reach the criterion in both phases compared with the NoShock/Veh and Shock/URB groups (for both groups: acquisition, *p* < 0.01; reversal, *p* < 0.001). This suggests that URB597 prevented the shock and reminders-induced impairment in performance in the WTM task.

#### 3.1.6. Forced Swim Test

For the FST, performed on days 30–31 (Figure 1f), a two-way ANOVA [shock × drug; 2 × 2] on immobility revealed significant effects of drug [F(1,36) = 6.047, *p* < 0.05], and drug × shock interaction [F(1,36) = 7.389, *p* < 0.05]. Post hoc analysis revealed an increase in immobility in the Shock/Veh group compared with the NoShock/Veh group (*p* < 0.05) and the Shock/URB597 group (*p* < 0.01), suggesting that URB597 restored the shock- and reminders-induced despair-like behavior.

#### 3.1.7. β-Catenin

Punches were extracted from the rat mPFC, NAc, CA1, and BLA (Figure 1g–j, respectively; also shown are brain sites from where the tissue samples were extracted). 

For β-catenin levels, a two-way ANOVA [shock × drug; 2 × 2] revealed significant effects of shock [NAc: F(1,36) = 8.52, *p* < 0.01], drug [mPFC: F(1,36) = 15.324, *p* < 0.001; NAc: F(1,36) = 24.099, *p* < 0.001; CA1: F(1,36) = 8.749, *p* < 0.01], and shock × drug interaction [mPFC: F(1,36) = 4.320, *p* < 0.05; NAc: F(1,36) = 16.888, *p* < 0.001; CA1: F(1,36) = 10.863, *p* < 0.01]. Post hoc comparisons revealed a significant decrease in β-catenin levels in the mPFC and NAc in the Shock/Veh group compared with the NoShock/Veh group (mPFC: *p* < 0.05; NAc: *p* < 0.01) and the Shock/URB597 group (mPFC: *p* < 0.01; NAc *p* < 0.001). In CA1, the Shock/URB597 group demonstrated increased β-catenin levels compared with the NoShock/URB597 group and the Shock/Veh group (both *p* < 0.01). No significant effects were observed in the BLA. Hence, exposure to shock and reminders downregulated β-catenin levels in the mPFC and NAc, and URB597 normalized these effects.

The same blots were rehybridized with antibodies specific for β-actin in order to confirm equal protein loading. As there were no differences between the groups in the levels of β-actin in the brain regions we examined, we concluded that the treatment had no effect on the levels of β-actin.

#### 3.1.8. Correlation between β-Catenin Levels and Behavior

We conducted Pearson bivariate correlation tests (Appendix A) between the expression of β-catenin and behavior to explore the association between the β-catenin levels and the anxiety- and depressive-like phenotype of the rats. The most robust correlations were found between β-catenin levels in the NAc and the following behaviors: freezing (SR1: r = −0.560; SR2: r = −0.550; SR3: r = −0.610; SR4: r = −0.572; SR1: r = −0.541; all *p* < 0.01); ASR (r = −0.461, *p* < 0.01); WTM acquisition (r = −0.437, *p* < 0.01) and reversal (r = −0.694, *p* < 0.01), and saccharin preference (Day 2: r = 0.618, *p* < 0.01; Day 7: r = 0.597, *p* < 0.01). This suggests that decreased β-catenin levels in the NAc are associated with enhanced freezing and startle response, anhedonia, and impaired performance in the WTM. A robust correlation was also found between mPFC β-catenin levels and climbing in the FST (r = 0.419, *p* < 0.01).

### 3.2. Experiment 2: The Preventing Effects of URB597 on Behavior of Rats Exposed to Shock and Reminders Are Mediated by CB1 Receptors

As URB597 is a FAAH inhibitor, we aimed to examine whether its effects on stress behavior are mediated through CB1r-dependent mechanisms. We used a low dose of the CB1r antagonist AM251 (am; 0.3 mg/kg) to block CB1r, as previous results have demonstrated that a low dose of this antagonist had no effect on behavior by itself but prevented the therapeutic effects of the cannabinoid agonists (Segev et al., 2018). Therefore, AM251 and URB597 were administered concurrently in order to examine the involvement of CB1rs in the effects of the FAAH inhibitor on behavior (for a detailed study design, see Section 2.13 Experimental Design).

#### 3.2.1. Freezing

Freezing behavior was monitored during the 1 min exposure to the SRs. A repeated measures ANOVA (shock × drug × SR; 2 × 3 × 6) revealed significant main effects of shock [F(1,42) = 551.682, *p* < 0.001] and SR [F(5,210) = 33.124, *p* < 0.001]; with a shock × SR interaction [F(5,210) = 32.356, *p* < 0.001] (Figure 2a). Post hoc analysis revealed that on SR1 to SR5, the shock groups (Veh; am; am + URB) demonstrated a significant increase in freezing levels compared with the NoShock groups (Veh; am; am + URB; *p* < 0.001). Hence, the preventive effect of URB597 on freezing behavior was blocked by AM251.

#### 3.2.2. Saccharin Preference

Saccharin preference was tested on day 2 pre-shock and on days 2, 7, and 14 post-shock. A repeated measures ANOVA (2 × 2 × 4) revealed a significant effect of shock [F(1,54) = 99.552, *p* < 0.001] and drug [F(2,54) = 6.626, *p* < 0.01], with a shock × drug interaction [F(1,54) = 4.021, *p* < 0.05] (Figure 2b). Post hoc analysis revealed that the shock groups (Veh, am, am + URB) demonstrated a decrease in saccharin preference compared with their corresponding NoShock groups: NoShock/Veh (days 2, 7, and 14: all *p* < 0.001), NoShock/am (day 2: *p* < 0.05; day 7: *p* < 0.01), NoShock/am + URB (day 2: *p* < 0.01; day 7: *p* < 0.001; days 14 and 21: *p*< 0.05). Moreover, compared with the NoShock/Veh group, we found that the NoShock/am group (days 2 and 14: *p* < 0.01; day 7: *p* < 0.001), and the NoShock/am + URB group (days 2 and 14: *p* < 0.05; day 7: *p* < 0.01) demonstrated decreased saccharin preference. Hence, co-administration of URB597 + AM251 shows that the restoring effect of URB597 on saccharin preference was blocked by AM251 treatment.

The shock-induced decrease in saccharin preference was short-termed (observed only one or two weeks after shock exposure). We have previously shown that in rats exposed to severe shock and to SRs 7, 14, and 21 days after shock exposure, the shock-induced decrease lasted till day 14 post-shock but not till day 28 post-shock (Burstein et al., 2018). In a previous study, we found that the chronic stress-induced decrease in sucrose consumption only lasted 1 week after the stress ended (Segev et al., 2014). Hence, there are fluctuations in saccharin preference at different times after stress exposure.

#### 3.2.3. Acoustic Startle Response

We tested ASR on days 1 and 26, and a repeated measures ANOVA on startle amplitude indicated significant main effects of shock [F(1,54) = 4.060, *p* < 0.001] and time [F(1,54) = 17.854, *p* < 0.001], as well as a shock × time interaction [F(1,54) = 17.854, *p* < 0.001] (Figure 2c). Post hoc analysis revealed that on ASR2, a significant increase in amplitude was found in the shocked groups treated with vehicle (Shock/Veh), AM251 alone (Shock/am), and AM251 and URB597 (Shock/am + URB597) compared with the non-shocked groups (NoShock/Veh: *p* < 0.01, NoShock/am: *p* < 0.01, NoShock/am + URB597: *p* < 0.001). Hence, AM251 had no effect on ASR by itself; the co-administration of URB597 and AM251 in rats exposed to shock and reminders increased their startle response amplitude, suggesting that the effects of URB597 on startle response were blocked by AM251 treatment.

#### 3.2.4. Social Tests

For the social tests performed on day 27, a two-way ANOVA revealed significant main effects of shock [preference: F(1,42) = 419.047, *p* < 0.001; recognition: F(1,42) = 315.307, *p* < 0.001] and drug [preference: F(2,42) = 9.021, *p* < 0.01], with a shock × drug interaction [preference: F(2,42) = 5.161, *p* < 0.05; recognition: F(2,42) = 6.969, *p* < 0.01] (Figure 2d). Post hoc analysis revealed a significant decrease in the exploration ratio in both tasks in the shocked groups (Shock/Veh, Shock/am, Shock/am + URB597) compared to the non-shocked groups (NoShock/Veh, NoShock/am, NoShock/am + URB597; all *p* < 0.001). Hence, AM251 had no effect on social behavior by itself; shocked rats co-administrated with URB597 and AM251 behaved similarly to shocked rats treated with vehicle, suggesting that the effects of URB597 on social behavior were blocked by AM251 treatment. For total exploration time, see Appendix A.

#### 3.2.5. Water T-Maze

Rats were tested in the WTM on days 28–29. Repeated measures ANOVA (2 × 2 × 2) revealed significant main effects of shock [F(1,54) = 85.307, *p* < 0.001], drug [F(2,54) = 5.788, *p* < 0.01], and time [F(1,54) = 9.179, *p* < 0.01]. We also detected two interactions: shock × drug [F(2,54) = 5.840, *p* < 0.01] and time × shock [F(1,54) = 6.950, *p* < 0.01] (Figure 2e). Post hoc analysis revealed that fewer trials were needed to reach the criterion in the NoShock groups (Veh, am, am + URB) compared with their corresponding shock groups: Shock/Veh (acquisition and reversal, both *p* < 0.001), Shock/am (reversal, *p* < 0.01), and Shock/am + URB (acquisition and reversal, both *p* < 0.01). In addition, the NoShock/Veh group needed fewer trials to reach criterion compared with the NoShock/am group (acquisition, *p* < 0.001; reversal, *p* < 0.01) and the NoShock/am + URB group (acquisition, *p* < 0.001; reversal, *p* = 0.05) groups. Hence, the co-administration of URB597 + AM251 shows that the restoring effect of URB597 on performance in this task was blocked by AM251 treatment.

#### 3.2.6. Forced Swim Test

For FST performed on days 30–31, a two-way ANOVA on immobility revealed a significant main effect of shock [F(1,42) = 70.807, *p* < 0.001] as well as a shock × drug interaction [F(1,42) = 3.660, *p* < 0.05] (Figure 2f). Post hoc analysis revealed that, compared to the non-shocked groups, the shocked groups demonstrated a significant increase in immobility (Shock/Veh, *p* < 0.01; Shock/am, Shock/am + URB597, *p* < 0.001). Hence, the effects of URB597 on immobility were blocked by AM251 treatment. 

No significant differences were observed between the Shock/Veh and Shock/am groups in any of the tests, suggesting that this low dose of AM251 had no effect on behavior by itself. We did detect differences between the Noshock/Veh and Noshock/am groups in the saccharin and WTM tests, suggesting an effect of AM251 in these tests in control rats. The fact that AM251 had an effect by itself in these two tests in non-stressed rats could suggest that under these conditions, the co-administration of URB with AM251 has additive effects (i.e., the combining effects of the two drugs equal the sum of the effects of the two drugs acting independently); hence, the effects of URB597 in WTM performance and social preference are not necessarily mediated by CB1r.

### 3.3. Experiment 3: The Effects of NAc β-Catenin Overexpression on Behavior in Rats Exposed to Shock and Reminders

We found that exposure to shock and reminders induced a behavioral phenotype that includes anxiety- and depressive-like behaviors, impaired memory performance, and decreased expression of β-catenin in the NAc and mPFC compared to non-shocked rats. Pearson correlations indicated that the behavioral phenotype was highly associated with decreased levels of β-catenin in the NAc. Hence, we next examined whether the overexpression (OE) of β-catenin in the NAc would prevent the effects of exposure to shock and reminders on behavior.

#### 3.3.1. Verifying β-Catenin Overexpression and Accuracy of Injection

In a preliminary experiment, we delivered overexpression (OE) vectors into the NAc (Figure 3a). In one set of rats (n = 12) we measured β-catenin expression in the NAc using WB (Figure 3b). An independent sample t-test revealed that overexpressing β-catenin in the NAc resulted in significant upregulation of β-catenin levels in the NAc 5 days after viral delivery [t(10) = 3.230, *p* < 0.05] compared to the GFP group. In the second set of rats (n = 12) we verified the accuracy of injection in the NAc using GFP detection (Figure 3c).

We examined whether viral-mediated OE of β-catenin in the NAc can restore the effects of shock and reminders on behavior, compared to rats injected with GFP (see Section 2.13 Experimental Design).

#### 3.3.2. Freezing

A repeated measures ANOVA (shock × virus × SR; 2 × 2 × 6) on freezing behavior during SRs (Figure 4a) indicated significant main effects of drug [F(1,30) = 27.535, *p* < 0.001], shock [F(1,30) = 56.779, *p* < 0.001], and SR [F(5,150) = 14.616, *p* < 0.001]. We also detected the following significant interactions: shock × virus [F(1,30) = 23.092, *p* < 0.001], shock × SR [F(5,150) = 7.636, *p* < 0.05], virus × SR [F(5,150) = 20.135, *p* < 0.001], and shock × virus × SR [F(5,150) = 12.028, *p* < 0.01]. Post hoc analysis revealed that on SR1–SR5, the Shock/GFP group demonstrated a significant increase in freezing levels compared with the NoShock/GFP group (SR1, *p* < 0.01; SR2–SR5, *p* < 0.001) and Shock/OE group (SR1, *p* < 0.05; SR2, *p* < 0.01, SR3–SR5: *p* < 0.001). Moreover, the Shock/OE group showed increased freezing levels compared with the NoShock/OE group (SR1, *p* < 0.01; SR2, SR3, SR5: *p* < 0.05). Taken together, the findings suggest that β-catenin OE ameliorated the shock- and reminders-induced increase in freezing behavior.

Next, we examined whether viral–mediated OE of β-catenin in the NAc can restore the effects of shock and reminders on behavior, compared to rats injected with GFP (see Section 2.13 Experimental Design). In this experiment, the brains were taken for analysis 26 days after virus delivery and following behavioral testing.

#### 3.3.3. Saccharin Preference

A mixed design three-way ANOVA on saccharin preference (shock × virus × days; 2 × 2 × 4) indicated significant main effects of shock [F(1,30) = 8.739, *p* < 0.01] and days [F(3,30) = 35.374, *p* < 0.001], with the following significant interactions: shock × virus [F(1,30) = 7.196, *p* < 0.05]; shock × days [F(1,30) = 7.90, *p* < 0.001]; and shock × virus × days [F(1,30) = 2.577, *p* = 0.05] (Figure 4b). Post hoc comparisons revealed that the Shock/GFP group demonstrated decreased saccharin preference compared with the NoShock/GFP group (day 2: *p* < 0.001) and the Shock/OE group (day 2: *p* < 0.001; day 14: *p* < 0.05). This suggests that NAc β-catenin overexpression prevented the shock- and reminders-induced decrease in saccharin preference.

#### 3.3.4. Acoustic Startle Response

A repeated measures ANOVA [shock × virus × time] on startle amplitude in the ASR indicated a significant main effect of virus [F(1,30) = 12.943, *p* < 0.01], and a significant shock × virus interaction [F(1,30) = 6.737, *p* < 0.05] (Figure 4c). Post hoc analysis revealed that the Shock/GFP group demonstrated a significant increase in startle amplitude compared with the Shock/OE and NoShock/GFP groups (both *p* < 0.001). Hence, upregulation of β-catenin in the NAc restored startle amplitude in rats exposed to shock and reminders.

#### 3.3.5. Social Tests

A two-way ANOVA for the social tests revealed significant main effects of shock [preference: F(1,30) = 17.652, *p* < 0.001; recognition: F(1,30) = 4.751, *p* < 0.05] and virus [preference: F(1,30) = 16.185, *p* < 0.001], with a significant shock × virus interaction [preference: F(1,30) = 8.402; recognition: F(1,30) = 8.727, both *p* < 0.01] (Figure 4d). Post hoc analysis revealed a lower exploration ratio in both tasks in the Shock/GFP group compared with the Shock/OE group (preference, *p* < 0.01; recognition, *p* < 0.001) and the NoShock/GFP group (preference, *p* < 0.001; recognition, *p* < 0.01). This suggests that NAc β-catenin overexpression prevented the effects of shock and reminders on social behaviors. For total exploration time, see Appendix A.

#### 3.3.6. Water T-Maze

A mixed design three-way ANOVA (shock × virus × test; 2 × 2 × 2) on the WTM experiment revealed significant main effects of shock [F(1,30) = 25.88, *p* < 0.001] and virus [F(1,30) = 23.67, *p* < 0.001]; with two significant interactions: shock × virus [F(1,30) = 28.735, *p* < 0.001] and test × shock × virus [F(1,30) = 7.50, *p* = 0.01] (Figure 4e). Post hoc comparisons revealed that the Shock/GFP group showed a significant increase in the number of trials required to reach the criterion in the acquisition and reversal phases compared with the NoShock/GFP group (acquisition, *p* < 0.01; reversal, *p* < 0.001) and Shock/OE group (acquisition and reversal, both *p* < 0.001). Hence, overexpression of β-catenin in the NAc prevented the shock- and reminders-induced impairment in performance in the WTM task.

#### 3.3.7. Forced Swim Test

Two-way ANOVA for the FST experiment revealed significant main effects on immobility of shock [F(1,30) = 78.646, *p* < 0.001] and virus [F(1,30) = 78.695, *p* < 0.001], with a significant shock × virus interaction [F(1,30) = 98.321, *p* < 0.01] (Figure 4f). Post hoc analysis revealed an increase in immobility in the Shock/GFP group compared with the NoShock/GFP and Shock/OE (both *p* < 0.001) groups, suggesting that overexpression of β-catenin in the NAc prevented the shock- and reminders-induced impairment in the FST.

No significant differences were observed between the NoShock/GFP and the NoShock/OE groups in any of the behavioral measures, suggesting that the virus had no effect in non-shocked control rats.

### 3.4. β-Catenin, mGluR5, and CB1 Receptors Regulation by Overexpressing NAc β-Catenin in Rats Exposed to Shock and Reminders

#### 3.4.1. β-Catenin 

Following the behavioral battery (Figure 4a) and the sacrifice of the rats, the expression of β-catenin was measured in the mPFC (Figure 5a), NAc (Figure 5b), CA1 (Figure 5c), and BLA (Figure 5d). A two-way ANOVA on β-catenin levels revealed significant main effects of shock [mPFC: F(1,30) = 13.278, *p* < 0.01; NAc: F(1,31) = 11.402, *p* < 0.01] and virus [BLA: F(1,31) = 11.593; CA1: F(1,31) = 9.944, all *p* < 0.01], as well as a shock × virus interaction [mPFC: F(1,30) = 13.781, *p* < 0.01; NAc: F(1,31) = 9.515, *p* < 0.01; CA1: F(1,31) = 19.697, *p* < 0.001]. Post hoc analysis revealed decreased β-catenin levels in the mPFC, NAc, and CA1 in the Shock/GFP group compared with the NoShock/GFP group (mPFC, *p* < 0.001; NAc and CA1, both *p* < 0.01) and the Shock/OE group (mPFC, *p* < 0.05; Nac and CA1, both *p* < 0.001). Additionally, in the CA1 the Shock/OE group demonstrated increased expression compared to the NoShock/OE group (*p* < 0.05), and in the BLA, the Shock/OE group demonstrated increased expression compared to the Shock/GFP group (*p* < 0.01). Hence, exposure to shock and reminders downregulated β-catenin levels in the mPFC, Nac, and CA1, and overexpressing β-catenin in the NAc normalized these effects. 

For β-actin levels, no significant between-group differences were observed in any of these brain areas, suggesting that β-actin levels were not affected by the treatment.

#### 3.4.2. mGluR5

The expression of mGluR5 was measured in the mPFC (Figure 5e), NAc (Figure 5f), CA1 (Figure 5g), and BLA (Figure 5h). A two-way ANOVA on mGluR5 protein levels revealed significant main effects of shock [NAc: F(1,31) = 13.209, *p* < 0.01] and virus [CA1: F(1,31) = 6.252, *p* < 0.05; mPFC: F(1,31) = 17.391, *p* < 0.001], as well as a shock × virus interaction [mPFC: F(1,31) = 11.468, *p* < 0.01; NAc: F(1,31) = 24.915, *p* < 0.001; CA1: F(1,31) = 4.682, *p* < 0.05]. Post hoc analysis revealed a decrease in mGluR5 levels in the mPFC and NAc in the Shock/GFP group compared with the NoShock/GFP group (mPFC: *p* < 0.01; NAc: *p* < 0.001) and the Shock/OE group (mPFC: *p* < 0.001; NAc: *p* < 0.01). Additionally, in the NAc the NoShock/GFP group demonstrated increased expression compared with the NoShock/OE group (*p* < 0.05), and in the CA1 the NoShock/GFP group demonstrated decreased expression compared with the NoShock/OE group (*p* < 0.01) and the Shock/GFP group (*p* < 0.05). No effects were observed in the BLA. Hence, exposure to shock and reminders downregulated mGluR5 levels in the mPFC and NAc, and overexpressed β-catenin in the NAc normalized these effects. OE decreased mGluR5 levels compared to the GFP group in the non-shocked groups. Nevertheless, OE restored the shock- and reminders-induced decrease in mGluR5 levels compared to GFP-shocked rats.

#### 3.4.3. CB1

Our findings regarding the involvement of β-catenin in the effects of shock and reminders on the amygdala-hippocampal-cortico-striatal circuit suggest a key role for the mPFC and the NAc in mediating the effects on behavior. Hence, we also examined the expression of CB1r in the mPFC and NAc. A two-way ANOVA on CB1r protein levels revealed significant main effects of shock [mPFC: F(1,31) = 14.405, *p* < 0.01; NAc: F(1,31) = 6.839, *p* < 0.05] and virus [mPFC:F(1,31) = 10.727, *p* < 0.01; NAc: F(1,31) = 4.030, *p* = 0.05], as well as a shock × virus interaction [mPFC: F(1,31) = 20.238, *p* < 0.001; NAc: F(1,31) = 24.915, *p* < 0.001]. Post hoc analysis revealed increased CB1r levels in the mPFC (Figure 5i) and NAc (Figure 5j) in the Shock/GFP group compared with the NoShock/GFP and Shock/OE groups (mPFC: both *p* < 0.001; NAc: both *p* < 0.01). Hence, exposure to shock and reminders upregulated CB1r levels in the mPFC and NAc, and overexpressing β-catenin in the NAc normalized these effects.

### 3.5. Experiment 4: The Effects of NAc β-Catenin Downregulation on Behavior of Rats Exposed to Shock and Reminders and Treated with URB597

#### 3.5.1. Verifying β-Catenin Downregulation 

In a preliminary experiment, we delivered an HSV P1005 vector, which expresses a mutant of β-catenin, into the NAc (Figure 3a). In one set of rats (n = 13) we measured β-catenin expression in the NAc using WB (Figure 3d). An independent sample t-test revealed that β-catenin protein levels in the NAc were downregulated (t(11) = 4.179, *p* < 0.01) compared to the HSV-GFP group. In another set of rats (n = 12), we verified the accuracy of injection in the NAc using GFP detection (Figure 3e).

Next, we examined whether the effects of URB597 on behavior in rats exposed to shock and reminders are mediated by β-catenin. To do so, we used a viral approach to downregulate (DR) β-catenin in the NAc (see Section 2.13 Experimental Design). 

#### 3.5.2. Freezing

For freezing during the SRs (Figure 6a), a repeated measures ANOVA (shock × virus × drug × SRs; 2 × 2 × 2 × 6) indicated significant main effects of drug [F(1,56) = 57.964, *p* < 0.001], shock [F(1,56) = 628.020, *p* < 0.001], virus [F(1,56) = 50.831, *p* < 0.001], and SR [F(5,280) = 21.893, *p* < 0.001]. The results also indicated the following interactions: shock × virus [F(1,56) = 52.261, *p* < 0.001], shock × drug [F(1,56) = 51.441, *p* < 0.001]; virus × drug [F(1,56) = 32.402, *p* < 0.001]; shock × virus × drug [F(1,56) = 41.557, *p* < 0.001]; shock × SR [F(5,280) = 20.870, *p* < 0.001], drug × SR [F(5,280) = 2.851, *p* < 0.05], shock × virus × SR [F(5,280) = 2.886, *p* < 0.05], and shock × virus × drug × SR [F(5,280) = 2.233, *p* = 0.05]. Post hoc analysis revealed that the Shock/GFP + Veh, Shock/DR + URB, and Shock/DR + Veh groups demonstrated a significant increase in freezing levels compared with their corresponding control groups (SR1–SR5: all *p* < 0.001). Moreover, the Shock/DR + URB group showed decreased freezing levels compared with the Shock/GFP + Veh group and the Shock/DR + URB group (SR1–SR5: all *p* < 0.001), and increased freezing levels compared with the NoShock/GFP + URB group (SR1, SR2, SR4, and SR5: all *p* < 0.05). Hence, downregulation of β-catenin in the NAc had no effect on freezing behavior by itself, but it blocked the preventive effects of URB597 in shocked rats.

#### 3.5.3. Saccharin Preference

For saccharin preference (Figure 6b), a repeated measures ANOVA (shock × virus × drug × time; 2 × 2 × 2 × 4) revealed significant main effects of shock [F(1,56) = 36.663, *p* < 0.001] and time [F(3,168) = 15.689, *p* < 0.001]. We also identified the following interactions: shock × virus [F(1,56) = 4.108, *p* < 0.05], shock × time [F(3,168) = 17.943, *p* < 0.001], and drug × time [F(3,168) = 3.263, *p* < 0.05]. Post hoc analysis revealed that on post-shock days 2 and 7, a significant decrease in saccharin preference was observed in the Shock/GFP + Veh group compared with the Shock/GFP + URB group (day 2: *p* < 0.05; day 7: *p* < 0.01) and the NoShock/GFP + Veh group (days 2 and 7: both *p* < 0.01). Additionally, a significant decrease was observed in the Shock groups (GFP + URB, DR + Veh, and DR + URB) compared their corresponding NoShock groups: Shock/GFP + URB (day 2: *p* < 0.01), Shock/DR + Veh (day 2: *p* < 0.01; day 7: *p* < 0.001), and Shock/DR + URB (day 2: *p* < 0.05; day 7: *p* < 0.01). Hence, downregulating β-catenin in the NAc had no effect on saccharin preference-shocked rats by itself; however, it blocked the restoring effects of URB597.

#### 3.5.4. Acoustic Startle Response

For ASR (Figure 6c), a repeated measures ANOVA revealed significant main effects of shock [F(1,56) = 108.120, *p* < 0.001], virus [F(1,56) = 19.008, *p* < 0.001], and drug [F(1,56) = 24.204, *p* < 0.001], as well as the following interactions: shock × virus [F(1,56) = 5.597, *p* < 0.05], shock × drug [F(1,56) = 34.653, *p* < 0.001], and shock × virus × drug [F(1,56) = 11.692, *p* < 0.01]. Post hoc analysis on ASR2 revealed decreased amplitude in shocked rats treated with URB597 (Shock/GFP + URB597) compared to the Shock/GFP + Veh and Shock/DR + URB597 groups (both *p* < 0.001). In addition, the shocked groups (Shock/GFP + Veh, Shock/DR + URB597, and Shock/DR + Veh) demonstrated increased startle compared to the non-shocked groups (NoShock/GFP + Veh, NoShock/DR + URB597, and NoShock/DR + Veh; all *p* < 0.001). The Shock/DR + URB597 group showed decreased amplitude compared to the Shock/DR + Veh group (*p* < 0.05). Hence, downregulating β-catenin in the NAc prevented the URB597 normalization of the startle response in shocked rats.

#### 3.5.5. Social Tests

For the social tests (Figure 6d), a three-way ANOVA revealed significant main effects of shock [preference: F(1,56) = 19.525; recognition: F(1,56) = 46.854; both *p* < 0.001], virus [preference: F(1,56) = 8.982, *p* < 0.01; recognition: F(1,56) = 4.958, *p* < 0.05], and drug [preference: F(1,56) = 4.008; recognition: F(1,56) = 4.447; both *p* < 0.05]. We also identified the following interactions: shock × drug [recognition: F(1,56) = 8.885, *p* < 0.01]; virus × drug [recognition: F(1,56) = 8.744, *p* < 0.01]; and shock × virus × drug [preference: F(1,56) = 15.375, *p* < 0.001]. Post hoc analysis revealed a significant increase in the exploration ratio in both tasks in the Shock/GFP + URB597 group compared to the Shock/GFP + Veh group (preference: *p* < 0.001; recognition: *p* < 0.01) and the Shock/DR + URB597 group (*p* < 0.01). This suggests that downregulation blocked the effects of URB597 on social behavior. 

In addition, we observed a decrease in the shocked groups (Shock/GFP + Veh, Shock/DR + URB597, Shock/DR + Veh) compared to the non-shocked groups (NoShock/GFP + Veh: preference, *p* < 0.05, recognition: *p* < 0.001; NoShock/DR + URB597: preference and recognition, *p* < 0.01; NoShock/DR + Veh: recognition, *p* < 0.01). In the preference task, an increase was observed in the NoShock/GFP + Veh group compared with NoShock/DR + Veh group (*p* < 0.05). Hence, NAc β-catenin downregulation blocked the preventive effects of URB597 on social behavior in rats exposed to shock and reminders. For total exploration time, see the Appendix A.

#### 3.5.6. Water T-Maze

For the WTM (Figure 6e), a repeated measures ANOVA (shock × virus × drug × time; 2 × 2 × 2 × 2) revealed significant main effects of shock [F(1,56) = 117.663, *p* < 0.001], virus [F(1,56) = 25.030, *p* < 0.001], and drug [F(1,56) = 13.504, *p* < 0.01]. We also identified the following significant interactions: shock × virus [F(1,56) = 19.996, *p* < 0.001], shock × drug [F(1,56) = 6.823, *p* < 0.05], virus × drug [F(1,56) = 7.294, *p* < 0.01], and shock × virus × drug [F(1,56) = 6.368, *p* < 0.05]. Post hoc analysis revealed that in the acquisition and the reversal phases, fewer trials were needed to reach the criterion in the non-shocked groups (GFP + Veh, DR + Veh, DR + URB) compared with the shocked groups: Shock/GFP + Veh (acquisition, *p* < 0.01; reversal, *p* < 0.001), Shock/DR + Veh (acquisition, *p* < 0.001; reversal: *p* < 0.01) and Shock/DR + URB (acquisition, *p* < 0.001; reversal, *p* < 0.01). The Shock/GFP + URB group also demonstrated a decreased number of trials compared with the Shock/GFP + Veh group (acquisition, *p* < 0.001; reversal, *p* < 0.01) and the Shock/DR + URB group (acquisition: *p* < 0.001; reversal: *p* < 0.01). Hence, downregulating β-catenin in the NAc had no effect on WTM performance, but it did block the preventative effects of URB597 in shocked rats.

#### 3.5.7. Forced Swim Test

For the FST (Figure 6f), a three-way ANOVA on immobility revealed significant main effects of shock [F(1,56) = 283.367, *p* < 0.001], virus [F(1,56) = 23.916, *p* < 0.001], and drug [F(1,56) = 23.916, *p* < 0.001], as well as the following significant interactions: shock × virus [F(1,56) = 30.606, *p* < 0.001], shock × drug [F(1,56) = 23.423, *p* < 0.001], virus × drug [F(1,56) = 24.432, *p* < 0.001], and shock × virus × drug [F(1,56) = 17.059, *p* < 0.001]. Post hoc analysis revealed that the Shock/GFP + URB597 group demonstrated decreased immobility compared with the Shock/GFP + Veh and Shock/DR + URB597 (both *p* < 0.001) groups. In addition, an increase in immobility was observed in the other shocked groups (Shock/GFP + Veh, Shock/DR + URB597, and Shock/DR + Veh) compared to the non-shocked counterparts (NoShock/GFP + Veh, NoShock/DR + URB597, and NoShock/DR + Veh; all *p* < 0.001). Hence, downregulating β-catenin in the NAc blocked the preventive effects of URB597 on despair-like behavior in the FST in rats exposed to shock and reminders.

No significant differences were observed between the Shock/GFP + Veh and Shock/DR + Veh groups, suggesting that downregulation had no effect on behavior by itself in shocked rats. In the non-shocked groups, we observed a difference between the NoShock/GFP + Veh and NoShock/DR + Veh groups in the social preference test, suggesting that downregulation decreased social preference.

## 4. Discussion

Our findings suggest a potentially novel mechanism for the stress-protective effects of URB597 through β-catenin activation in the NAc in a rat model for PTSD and depression. 

We found that exposing rats to shock and reminders induced anxiety-like behavior (i.e., increased freezing and startle response) and depressive-like behavior (i.e., decreased social behavior, induced anhedonia, and despair-like behavior), and also impaired memory function in the social recognition and water maze tasks. Administering the FAAH inhibitor URB597 1 h after shock exposure prevented these effects, as had been previously demonstrated [3,17]. Most of the effects of URB597 on behavior were found to be CB1r-dependent, as co-administration of the CB1r antagonist AM251 with URB597 prevented the ameliorating effects of URB597. 

Importantly, exposure to shock and reminders decreased β-catenin levels in the NAc and mPFC. Decreased β-catenin levels in the NAc were associated with the behavioral phenotype, including enhanced freezing and startle response, anhedonia, and impaired performance in the WTM. This association led us to focus on upregulating and downregulating β-catenin levels in the NAc. Overexpression of NAc β-catenin in rats exposed to shock and reminders resulted in intact behavior (restores freezing and startle response, no indication of depressive-like behavior, intact memory function) and restored expression of β-catenin, mGluR5, and CB1r in the NAc. It is interesting that overexpressing β-catenin in the NAc also restored these same normal levels in the mPFC. Future studies might test the overexpression of β-catenin in the mPFC. Another future study is to examine the potential therapeutic effects of URB597 and the involvement of β-catenin in female rats.

Increased β-catenin levels were reported to promote resilient responses to stress in the NAc [26]. Mice with stabilized β-catenin in the hippocampus showed resilience to some anxious/depressive manifestations when subjected to the corticosterone model of depression [49]. NAc β-catenin upregulation in mice exposed to social defeat stress resulted in a pro-resilient phenotype, demonstrating less social avoidance and better performance in the FST and elevated plus maze (Dias et al., 2014). Similarly, intra-NAc LiCl (2 µg/side), which upregulates β-catenin activity via inhibition of GSK-3β, facilitated inhibitory extinction [26]. 

In rats exposed to shock and reminders, we found that β-catenin overexpression restored the decrease in mGluR5 and the increase in CB1r expression in the NAc and mPFC. Decreased mGluR5 levels were observed in animal models of depression and the mPFC of depressed individuals [34,35], and blocking mGluR5 has therapeutic effects in PTSD patients and animal models [34,50]. Increased CB1r in the BLA and CA1 were observed in animal models for PTSD [4,15,17], and increased CB1r availability in the amygdala-hippocampal-cortico-striatal circuit in human subjects with PTSD [10]. 

Other studies have shown a stress-induced decrease in β-catenin levels in the mPFC and NAc that was accompanied by highly susceptible behavioral responses [22,51,52,53]. Mice with dysfunctional NAc Wnt signaling demonstrated increased depression-like behavior and susceptibility to social defeat stress [23], and β-catenin inactivation in the astrocyte-specific glutamate transporter (GLAST)-expressing cells enhanced anxious/depressive-like responses [49]. In depressed patients, lowered β-catenin protein levels, but not mRNA levels were found in the NAc, suggesting that depression may be associated with reduced activity of β-catenin, and perhaps not a defect at the transcriptional level [49,54]. 

Importantly, URB597 did not prevent the effects of exposure to shock and reminders on behavior in rats with viral-mediated NAc downregulation of β-catenin. This indicates that β-catenin is crucial for URB597 to exert its ameliorating effects on behavior. In a previous study, we showed that downregulating β-catenin using sulindac prevented the facilitating effect of the CB1/2 agonist WIN55,212-2 on extinction [26]. Taken together, these findings suggest a strong functional interaction between CB1r and β-catenin. Indeed, cannabinoids regulate neuronal precursor proliferation via β-catenin; the activation of CB1r enhances the activity of PI3K/AKT; this results in AKT-mediated phosphorylation of GSK-3β, followed by the stabilization of β-catenin that translocates into the nucleus; in the nucleus, β-catenin regulates transcription and gene expression such as cyclin D1 that is involved in cell proliferation regulation [55]. β-catenin may activate TCF/Lef transcription factors [56] and microRNAs [57], which promote anti-stress responses. This could be a possible explanation for the therapeutic-like effects of URB597, acting through CB1r to modulate β-catenin and produce pro-resilient responses. 

There are other relevant pathways that might play a critical role in explaining our results. For example, AEA affects CB1r but also has other targets that might be involved in the effects of URB597 [i.e., AEA is a full agonist of TRPV1, which probably participates in ECB signaling [58]]. Garro-Martinez et al. (2020) suggested a link between β-catenin levels and 5-HT1A receptor functionality underlying the vulnerability or resilience to stress-related disorders [59].

## 5. Conclusions

In many cases, drugs are used in clinical settings without a full understanding of the molecular mechanisms through which they function. Understanding the mechanism of action for a given drug in greater detail has the potential to support further pharmacological development efforts and mitigate the risk of failed clinical trials by stratifying patients to focus on subpopulations most likely to respond to such treatment. We suggest a potentially novel mechanism for the stress-ameliorating effects of URB597 that involves activation of CB1 and the Wnt/β-catenin pathway in the NAc. Overall our findings suggest that FAAH inhibitors may be a viable approach for the treatment of stress-related neuropsychiatric disorders and PTSD in particular and that these therapeutic effects are mediated via a β-catenin-dependent mechanism. 

## Data Availability

Not applicable.

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
