# Peer review of "Enhancing Endocannabinoid Signaling via β-Catenin in the Nucleus Accumbens Attenuates PTSD- and Depression-like Behavior of Male Rats"

_biomedicines, 2022, doi:10.3390/biomedicines10081789_

Round 1

Reviewer 1 Report

The manuscript focuses on understanding the mechanism by which the FAAH inhibitor URB597 regulates stress related conditions i the nucleus accumbens. The study is overall well designed with adequate power and follows a generally expected sequence of assays to address the role of b-catenin after URB597 treatment. The methods and statistics are comprehensive. Throughout only male rates are utilized and therefore sex is not considered as a biological variable without proper rationale. Concentrations of drugs are developed from previous literature. Below are some major and minor points associated with my review.

Is there a statistical significance between each SR reminder (1-5)? As it appears measurement was taken only after SR5 (minor)

The 14 day saccharine time point is not changed after shock, but not explained in Figure 1. It is different in subsequent figures. This should be discussed (major)

The figures for social preference in figure 1 and 2 are mixed up (minor)

For some experiments using AM251 with or without the drug of interest has an effect on the animal behavior (water maze test and forced swim test) but not discussed. It raises questions as to whether these experiments can be interpreted. While not the case in all behavioral assays, the changes are relatively ignored and not discussed (major)

Control expression of b-catenin in the OE and DR assays is not addressed by IHC. The b-catenin is not very high over background and not convincing (major)

In the intro or lead up to the mGluR experiments there is a lack of rationale for the experiments (moderate)

Figure 5 has a major flaw the No shock OE group appears to have less b-catenin or equal expression than the no shock gfp. That is not what the validation studies in the previous figure show. The expression is further decreased by shock in GFP which is consistent with conclusions. But without the increase in the b-catenin in the OE control the experiments are somewhat not interpretable. Except for in the CA1 and the BLA (major)

The knockdown experiments in figure 6 are highly convincing but since the previous validation of the DR lacks a control part it is less strong.

Author Response

We thank the reviewer for the valuable comments. Our corrections are detailed below.

Reviewer 1

Comments and Suggestions for Authors

The manuscript focuses on understanding the mechanism by which the FAAH inhibitor URB597 regulates stress related conditions i the nucleus accumbens. The study is overall well designed with adequate power and follows a generally expected sequence of assays to address the role of b-catenin after URB597 treatment. The methods and statistics are comprehensive. Throughout only male rates are utilized and therefore sex is not considered as a biological variable without proper rationale. Concentrations of drugs are developed from previous literature. Below are some major and minor points associated with my review.

We have started a project with females exposed to the shock and reminders model for PTSD and have promising results with the pharmacology and behaviour. We plan to further study the underlying mechanisms.

A comment regarding this (Line 901): Another future study is to examine the potential therapeutic effects of URB597 and the involvement of β-catenin in female rats.

Is there a statistical significance between each SR reminder (1-5)? As it appears measurement was taken only after SR5 (minor)

The measurements were taken after SR5 in order to examine the long-term effects of the traumatic event.

It has become increasingly clear that the consequences of exposure to trauma are affected not only by aspects of the event itself, but also by the frequency and severity of trauma reminders (Korem and Akirav, 2014). Places and situations are the most frequent trauma reminders (Pynoos et al 1996; Louvart et al 2005; Corral-Frias et al, 2013). In a previous study we compared rats that were exposed to the shock followed by two situational reminders (SRs) with rats that were exposed only to the shock (with no SRs).  Shocked rats exposed to SRs persistently avoided the dark chamber, whereas, shocked rats not exposed to SRs demonstrated increased avoidance on the first extinction trial, but their extinction kinetics was intact (Korem and Akirav, 2014). In more recent studies we applied more exposures to SRs in order to examine the long-term effects of the traumatic event (Burstein et al., 2018; Shoshan et al., 2017).

We added the following explanation (Line 135): Behavioral testing was taken after SR5 in order to examine the long-term effects of the traumatic event. In a previous study we found that shocked rats exposed to SRs persistently avoided the dark chamber, whereas, shocked rats not exposed to SRs demonstrated increased avoidance on the first extinction trial, but their extinction kinetics was intact (Korem and Akirav, 2014).

The 14 day saccharine time point is not changed after shock, but not explained in Figure 1. It is different in subsequent figures. This should be discussed (major)

We added the following explanation (Line 460): The shock-induced decrease in saccharine preference was short-termed (observed only one or two weeks after shock exposure). We have previously shown that in rats exposed to severe shock and to SRs 7, 14 and 21 days after shock exposure, the shock-induced decrease lasted till day 14 post-shock but not till day 28 post-shock (Burstein et al., 2018). In a previous study we found that chronic stress-induced decrease in sucrose consumption only lasted 1 week after the stress ended (Segev et al., 2014). Hence, there are fluctuations in saccharine preference at different times after stress exposure.

The figures for social preference in figure 1 and 2 are mixed up (minor)

Corrected

For some experiments using AM251 with or without the drug of interest has an effect on the animal behavior (water maze test and forced swim test) but not discussed. It raises questions as to whether these experiments can be interpreted. While not the case in all behavioral assays, the changes are relatively ignored and not discussed (major)

We added the following explanation (line 517): The fact that AM251 had an effect by itself in these two tests in non-stressed rats could suggest that under these conditions, co-administration of URB with AM251 has additive effects (i.e., the combining effects of the two drugs equal the sum of the effects of the two drugs acting independently); hence, the effects of URB597 in WTM performance and social preference are not necessarily mediated by CB1r.

Control expression of b-catenin in the OE and DR assays is not addressed by IHC. The b-catenin is not very high over background and not convincing (major)

We corrected the figure and explained it more clearly (see the new figure 3 and the updated figure legend and methods). We used western blotting to show quantitatively that the overexpression (OE) group demonstrated increased β-catenin levels compared to the Green fluorescent protein (GFP) group (figure 3b) and that the downregulation (DR) group demonstrated decreased β-catenin levels compared to the GFP group (figure 3d).  We also showed that GFP detection revealed successful delivery of the vector to the nucleus accumbens (NAc) (figures 3c, 3e).

Also, we placed representative blots of the overexpression and downregulation groups (Figure 3b, 3d).

In the intro or lead up to the mGluR experiments there is a lack of rationale for the experiments (moderate)

We added the following to the rational for the mGulR5 experiments (Line 69):

In vivo and in vitro evidence suggests that Wnt/ β-catenin signaling is downstream of metabotropic glutamate receptor subtype 5 (mGluR5); hence, mGluR5 signaling plays a key role in controlling neuronal gene expression by regulating the assembly of the N-cadherin/ β-catenin complex and consequently the expression of REST/NRSF (Repressor element 1-silencing transcription factor/neuron-restrictive silencer factor) in primary corticostriatal neurons (de Souza et al., 2020). 

Importantly, mGluR5 is implicated in the pathophysiology of several psychiatric disorders (Nicoletti et al., 2011). The direction of mGluR5 modulation that elicits antidepres-sant/anxiolytic-like effects has been inconsistent across studies (Li et al., 2005; Tatarczyńska et al., 2001; Xu et al., 2009). However, the PFC of postmortem brains from major depression disorder patients shows reduced mGluR5 protein expression (Deschwanden et al., 2011); in rodents, an essential role for mGluR5 in the NAc was found in promoting stress resilience, suggesting that a deficit in mGluR5-mediated signaling in this region may represent an endophenotype for stress-induced depression (Shin et al., 2015).

In a recent study, NAc mGluR5 activation ameliorated the effects of stress on depression-like behavior and pain, through ECB mediation, suggesting an association between ECB signaling and the expression of mGluR5 in stressed rats (Xu et al., 2021).

Figure 5 has a major flaw the No shock OE group appears to have less b-catenin or equal expression than the no shock gfp. That is not what the validation studies in the previous figure show. The expression is further decreased by shock in GFP which is consistent with conclusions. But without the increase in the b-catenin in the OE control the experiments are somewhat not interpretable. Except for in the CA1 and the BLA (major)

In this experiment the brains were taken for analysis 26 days after virus delivery and following behavioral testing. We added this sentence for clarity (Line 603).

As described in the methods section, the vector is expressed in vivo within 2-3 h, with maximal expression from 3-5 days post-injection that lasts only 8 days in vivo (Kim et al., 2016; Neve et al., 2005; Wang et al., 2016; Wilkinson et al., 2011).

In figure 3, the viral delivery was 5 days before analysis, demonstrating increased expression of β-catenin.

The knockdown experiments in figure 6 are highly convincing but since the previous validation of the DR lacks a control part it is less strong.

We  performed quantitative analysis, using western blots, to demonstrate that the expression of β-catenin is indeed decreased following DR delivery (see the new figure 3d).

Reviewer 2 Report

The aim of this study is to examine the role of β-catenin in the stress-attenuating effects of FAAH inhibition and to assess its function in anxiety, and depression-like behaviors as well as memory function.

The experiments are well designed and data is well analyzed.

Author Response

Thank you !

Reviewer 3 Report

The manuscript raises the important issue of a new therapeutic mechanism of β-catenin activation by inhibiting fatty acid amide hydrolase, which in the experimental model gives an antidepressant effect and alleviates stress. Authors, using the animal model of post-traumatic stress disorder (PTSD), investigate the effect of URB597 (FAAH inhibitor) on anxiety – and depression-like behaviours and memory function. They also tested whether observed effects are mediated via β-catenin and the involvement of the CB1 and mGluR5 receptors in the nucleus accumbens. 

This study is interesting and carefully executed approaching the problem through diverse perspectives, such as evaluating behavioral, molecular and functional aspects. This interesting manuscript is broad in content and of substantial interest to the readership.

The manuscript is well written and demonstrates a comprehensive set of data. However, some critical points remain regarding data analysis and presentation of the data. To further strengthen the manuscript, my suggestions are: 

Major points:

The chapter "materials and methods" requires the greatest correction.

1.     2.1 - Did the animals have an enrichment of the environment in the cages?

2.     I suggest putting "experimental design" as subsection 2.3. A graphical experimental timeline of the research process would be very good. Describe also how many animals took part in each experiment, how numerous the groups were, and whether any results were excluded. Include this sequence of behavioural tests in the methodology that was performed. Now WMT is after the FST.

3.     2.6, ll 136 and 142 – what was the “novel object”? 

4.     2.8 - Specify what was counted in this test?

5.     2.9 - Were the rats in single cages? If so, write it in 2.1

6.     2.10 - Add a tissue sampling graph here with atlas coordinates. Specify what part of the mPFC was taken (prelimbic, infralimbic, cingulate). How much protein was applied to the well and how many samples from each group were on each gel?

7.     2.11, l 209 - Are these coordinates relative to the Bregma point?

8.     2.12 - What anti- β-catenin antibody was used? Enter the number and manufacturer. Describe how the image analysis was performed.

9.     The description of the statistical analysis should be included in the next subsection (e.g. 2.13). What software was used for the statistical analysis? Has the normality of the distribution been checked and how?

10.  Was the locomotor activity of the animals assessed? For both memory tests and anxiety / depression tests, such an evaluation should be performed to rule out false-positive results.

Results:

1.     The presented graphs are very small and difficult to assess. Reduce the gaps between them and enlarge the graphs. 

2.     In all graphs a legend is missing regarding the data and the asterisk shown (e.g. "data is shown as mean +/- SEM, * p<0.05 compared to control group")

3.     Complete the captions under the figures so that they can be used without reading the entire chapter.

4.     Move the methodological descriptions of the "study design" to the methodology. Leave only the results in the results section.

5.     The results of the immunohistochemical analysis are missing.

6.     There are no individual points in the figures of saccharin measure.

7.     Line 580 – “The effect of NAc β-catenin overexpression on the expression of β-catenin” is unfortunate phrase. Use another. 

Discussion:

1.     Line 794 – “Increased β-catenin levels were reported to promote resilient responses to stress...” and “ ll 795-796 “Mice with stabilized β-catenin showed resilience to some anxious/depressive manifestations when subjected to the corticosterone model of depression” – specify brain structures.

2.     Emphasize the novelty of this project.

In all graphs a legend is missing regarding the data and the asterisk shown (e.g. "data is shown as mean  sem, * p<0.05 compared to control group"…).

Minor:

1.     ll 122 – correct the time format

2.     2.11 - Here, explain the HSV and GFP abbreviations.

3.     Fig 1d, 2d, 4d, 6d – complete the unit 

4.     Move the significance stars above the individual points

5.     line 548 – Correct to “Two-way ANOVA” (Capital letter)

It's a pity that the project only applies to males, while females are twice as likely to develop PTSD. Maybe future research will compare the effects in both sexes.

Author Response

We thank the reviewer for the valuable comments. Our corrections are detailed below.

Reviewer 3

The manuscript raises the important issue of a new therapeutic mechanism of β-catenin activation by inhibiting fatty acid amide hydrolase, which in the experimental model gives an antidepressant effect and alleviates stress. Authors, using the animal model of post-traumatic stress disorder (PTSD), investigate the effect of URB597 (FAAH inhibitor) on anxiety – and depression-like behaviours and memory function. They also tested whether observed effects are mediated via β-catenin and the involvement of the CB1 and mGluR5 receptors in the nucleus accumbens. 

This study is interesting and carefully executed approaching the problem through diverse perspectives, such as evaluating behavioral, molecular and functional aspects. This interesting manuscript is broad in content and of substantial interest to the readership.

The manuscript is well written and demonstrates a comprehensive set of data. However, some critical points remain regarding data analysis and presentation of the data. To further strengthen the manuscript, my suggestions are: 

Major points:

The chapter "materials and methods" requires the greatest correction.

  1. 1 - Did the animals have an enrichment of the environment in the cages?

We added the following (Line 104): Plastic hoses were placed in each cage to enrich the animals` environment.

I suggest putting "experimental design" as subsection 2.3. A graphical experimental timeline of the research process would be very good. Describe also how many animals took part in each experiment, how numerous the groups were, and whether any results were excluded. Include this sequence of behavioural tests in the methodology that was performed. Now WMT is after the FST.

Corrected

  1. 2.6, ll 136 and 142 – what was the “novel object”? 

We added (line 158) that the objects were children’s Lego blocks.

  1. 2.8 - Specify what was counted in this test?

We added that in both phases, the number of correct trials was recorded until rats reached 5 consecutive correct trials (Line 183).

  1. 2.9 - Were the rats in single cages? If so, write it in 2.1

We added in 2.1 (Line 105): For the saccharine preference test each rat was placed in a separate cage.

  1. 2.10 - Add a tissue sampling graph here with atlas coordinates. Specify what part of the mPFC was taken (prelimbic, infralimbic, cingulate). How much protein was applied to the well and how many samples from each group were on each gel?

Line 208: we added the atlas coordinates (coordinates relative to Bregma in mm: mPFC: anteroposterior (AP), +3.72; medial lateral (ML), ±0.4; ventral (V), -4.8; NAc shell: AP, +1.6; ML, ±1; V, -5.5; BLA: AP, -1.596; ML, ±4.2; V, 8.45; CA1: AP, -4.2; ML, ±2.5; V: -2.5).

Rat brain atlas illustrations indicating punch location are shown in Figure 1.

We added that the mPFC includes the prelimbic and infralimbic cortex (Line 206).

Wells were loaded with 30 μl of samples (Line 213).

We added that each gel contained at least one sample from each group (Line 214).

  1. 2.11, l 209 - Are these coordinates relative to the Bregma point?

We added that the coordinates are relative to Bregma.

  1. 2.12 - What anti- β-catenin antibody was used? Enter the number and manufacturer. Describe how the image analysis was performed.

We changed figure 3. We show that the expression of β-catenin is increased (with overexpression) or decreased (with downregulation) using western blotting. We also show, using IHC, that the GFP detection revealed successful delivery of the overexpressing vector to the nucleus accumbens. The methods were changed accordingly (see section 2.12).

  1. The description of the statistical analysis should be included in the next subsection (e.g. 2.13). What software was used for the statistical analysis? Has the normality of the distribution been checked and how?

We added the following: 2.14. Statistical analysis

Data were analyzed using SPSS 27 (IBM, Chicasgo, Illinois). Normality assumption was examined using Kolmogorov-Smimov and Shapiro-wilk tests.

  1. Was the locomotor activity of the animals assessed? For both memory tests and anxiety / depression tests, such an evaluation should be performed to rule out false-positive results.

 Locomotor activity was not assessed in these rats.

Results:

  1. The presented graphs are very small and difficult to assess. Reduce the gaps between them and enlarge the graphs. 

Corrected.

  1. In all graphs a legend is missing regarding the data and the asterisk shown (e.g. "data is shown as mean +/- SEM, * p<0.05 compared to control group")

Corrected.

  1. Complete the captions under the figures so that they can be used without reading the entire chapter.

Corrected.

  1. Move the methodological descriptions of the "study design" to the methodology. Leave only the results in the results section.

Corrected.

  1. The results of the immunohistochemical analysis are missing.

We changed figure 3 and corrected the methods section accordingly.

  1. There are no individual points in the figures of saccharin measure.

Corrected.

  1. Line 580 – “The effect of NAc β-catenin overexpression on the expression of β-catenin” is unfortunate phrase. Use another. 

Corrected (Line 673): β-catenin, mGluR5, and CB1 receptors regulation by overexpressing NAc β-catenin in rats exposed to shock and reminders

 Discussion:

  1. Line 794 – “Increased β-catenin levels were reported to promote resilient responses to stress...” and “ ll 795-796 “Mice with stabilized β-catenin showed resilience to some anxious/depressive manifestations when subjected to the corticosterone model of depression” – specify brain structures.

Corrected (Line 903): Increased β-catenin levels were reported to promote resilient responses to stress in the NAc (Korem et al., 2017). Mice with stabilized β-catenin in the hippocampus showed resilience to some anxious/depressive manifestations when subjected to the corticosterone model of depression (Vidal et al., 2019).

  1. Emphasize the novelty of this project.

We added the following at the beginning of the discussion section (Line 882): Our findings suggest a potential novel mechanism for the stress-protective effects of URB597 through β-catenin activation in the NAc in a rat model for PTSD and depression.

And the following to the conclusions (Line 953): In many cases, drugs are used in clinical settings without a full understanding of the molecular mechanisms through which they function. Understanding the mechanism of action for a given drug in greater detail has the potential to support further pharmacological development efforts and to mitigate the risk of failed clinical trials by stratifying patients to focus on subpopulations most likely to respond to such treatment. We suggest a potential novel mechanism for the stress-ameliorating effects of URB597 that involves ac-tivation of CB1 and the Wnt/β-catenin pathway in the NAc.  

  1. In all graphs a legend is missing regarding the data and the asterisk shown (e.g. "data is shown as mean ± sem, * p<0.05 compared to control group"…).

Corrected

Minor:

  1. ll 122 – correct the time format

Corrected

  1. 2.11 - Here, explain the HSV and GFP abbreviations.

Corrected

  1. Fig 1d, 2d, 4d, 6d – complete the unit 

Corrected to ASR amplitude (mV) (now figures 1c, 2c, 4c, 6c).

  1. Move the significance stars above the individual points

Corrected.

  1. line 548 – Correct to “Two-way ANOVA” (Capital letter)

Corrected.

It's a pity that the project only applies to males, while females are twice as likely to develop PTSD. Maybe future research will compare the effects in both sexes.

That is true. We have started a project with females and this shock and reminders model for PTSD and have promising results with the pharmacology and behaviour. We plan to further study the underlying mechanisms.

A comment regarding this in the discussion (Line 901): Another future study is to examine the potential therapeutic effects of URB597 and the involvement of β-catenin in female rats.

Round 2

Reviewer 1 Report

Most of the written comments as to rationale and experimental design, statistics were addressed by citations. 

However, there is still a major problem with Figure 3. The authors removed or it appears will remove the IHC of b-catenin, which was in question in the previous submission. By removing, they will show only the western blot quantification and appears to have removed localization of the NAc. In doing so this:
1) Brought attention to the representative western blots. The exact same blot is used for GFP injection in Figure 3b and in the DR knockdown in Figure 3d. In addition, the previous figure had N=7 in the DR sample and the new one has N=6. In addition, the GFP only from the DR group in Fig. 3d (new) has higher levels of expression and protein than the other groups. It is unclear what the basal level is and these lanes are cropped from the larger western blots so it is unclear on whether the samples have equal exposure or performed on different days. The most concerning is that the same western representative is being utilized for two independent samples (GFP in Fig. 3b and DR in Fig 3d). Based on these concerns, the assays using DR and OE cannot be interpreted.
2) The larger image of the NAc injection also raised the question as to the location and size of injection. By comparing the old and new figure 3, it appears that the injection bolus is very large and perhaps non-specific. More specifically, from old figure 3f, the NAc localization in the cartoon is closer to the white matter tract. The injection in Fig. 3f would therefore not be in the correct position. This would raise serious concerns as to the major conclusions of the paper as they focus on b-catenin NAc.

Author Response

Most of the written comments as to rationale and experimental design, statistics were addressed by citations. 

However, there is still a major problem with Figure 3. The authors removed or it appears will remove the IHC of b-catenin, which was in question in the previous submission. By removing, they will show only the western blot quantification and appears to have removed localization of the NAc. In doing so this:
1) Brought attention to the representative western blots. The exact same blot is used for GFP injection in Figure 3b and in the DR knockdown in Figure 3d. In addition, the previous figure had N=7 in the DR sample and the new one has N=6. In addition, the GFP only from the DR group in Fig. 3d (new) has higher levels of expression and protein than the other groups. It is unclear what the basal level is and these lanes are cropped from the larger western blots so it is unclear on whether the samples have equal exposure or performed on different days. The most concerning is that the same western representative is being utilized for two independent samples (GFP in Fig. 3b and DR in Fig 3d). Based on these concerns, the assays using DR and OE cannot be interpreted.

We added the representative blots for the different experiments (not cropped) (figures 3f and 3g).

We corrected the number of rats per group.

2) The larger image of the NAc injection also raised the question as to the location and size of injection. By comparing the old and new figure 3, it appears that the injection bolus is very large and perhaps non-specific. More specifically, from old figure 3f, the NAc localization in the cartoon is closer to the white matter tract. The injection in Fig. 3f would therefore not be in the correct position. This would raise serious concerns as to the major conclusions of the paper as they focus on b-catenin NAc.

We added bilateral expression of GFP in the NAc area (figure 3h) demonstrating that the injection was mostly localized to the nucleus accumbens.

Round 3

Reviewer 1 Report

New additions have addressed previous concerns.